# Study on Long Term Property of Soft Soil Solidified with Industrial Waste Residue and Regenerated Fine Aggregate

**DOI:** 10.3390/ma16062447

**Published:** 2023-03-19

**Authors:** Anhui Wang, Wanying Dong, Qiwei Zhan, Juanlan Zhou

**Affiliations:** 1China Construction Industrial & Energy Engineering Group Co., Ltd., Nanjing 210046, China; 2College of Transportation, Southeast University, Nanjing 211189, China; 3School of Civil Engineering and Architecture, Jiangsu University of Science and Technology, Zhenjiang 212003, China

**Keywords:** long-term property, industrial waste residue, recycled fine aggregate, soft soil

## Abstract

The long-term properties of solidified soft soil, including an immersion test, the dry–wet cycle and the freeze–thaw cycle, were systematically studied. Firstly, the immersion stability of solidified soft soil was confirmed. The appearance of soft soil solidified by a solidified agent and raw fine aggregate did not change significantly, and it was still intact without damage when the soaking time increased up to 28 d. Secondly, the mass and compressive strength loss of solidified soft soil were determined. When the number of dry–wet cycles was one, three, five and seven, the accumulated-mass loss rate was 1.4%, 3.0%, 4.5% and 6.0%, respectively, and the compressive-strength loss rate was −10.3%, 13.9%, 41.2% and 53.6%, respectively. Compared with solidified soft soil under standard curing environments, solidified soft soil after seven dry–wet cycles showed small cracks, and the structural compactness began to decline. Finally, the influence of the freeze–thaw cycle on the mass, compressive strength and microstructure of solidified soft soil was confirmed. When the number of freeze–thaw cycles was 5, 10, 15 and 20, the accumulated-mass loss rate was 12.6%, 16.7%, 17.9% and 18.8%, respectively. The microstructure of the solidified soft soil was damaged, and the increase in porosity was the main reason for its strength reduction or even failure. Nevertheless, soft soil with a solidified agent and recycled fine aggregate had no obvious damage to the microstructure, and the freeze–thaw resistance was relatively superior.

## 1. Introduction

Soft soil is a kind of engineering waste soil, which is characterized by its high water content, high clay content, poor water permeability, slow drainage consolidation, large compressibility and low bearing capacity [1,2]. Soft soil is difficult to be directly used as a filling material in engineering, and it is usually disposed of by throwing mud. The massive stacking of soft soil would not only occupy a lot of land resources but also pose a serious threat to the ecological environment. Meanwhile, soft soil could not be used directly, so a large amount of plain soil was purchased to meet the needs of the project, which greatly increased the construction cost. According to the disposal goals of stabilization, harmlessness and resource utilization, soft soil was transformed into renewable engineering materials by solidification technology, which is the effective way to solve the problem of soft soil [3,4,5,6,7].

In recent years, researchers have carried out systematic research on soft soil consolidation, and important progress has been achieved. Zhu et al. studied the influence of cement content and age on the mechanical properties of solidified soft soil, and the results showed that the unconfined compressive strength of solidified soft soil was linear with the cement content, while the failure strain decreased exponentially with the cement content; in addition, with an increase in the cement content and age, the stress–strain relationship of solidified soft soil changed from plasticity to brittleness [8,9,10]. Wang et al. found that increasing the cement content could improve the strength and durability of solidified soft soil to a certain extent; however, the durability of solidified soft soil made it difficult to achieve the desired effect only by adding cement [11]. Based on the traditional cement and lime solidification method, Wang et al. proposed a method of using a large amount of low-calcium fly ash, cement and lime to solidify the soft soil, in order to improve the solidification strength and durability; at the same time, the durability evolution model of active MgO and fly-ash-solidified soft soil was creatively obtained, which was helpful to deeply understand and clarify the durability evolution law under the interference of external environment [12,13]. Li et al. showed that the dry and wet durability of solidified sludge could be improved only when the cement content reached a certain level [14]. Cao et al. analyzed the microstructure of soft soil before and after solidification by scanning electron microscope; the porosity of solidified soft soil was significantly reduced, and the compactness was greatly improved, which was an important reason for the increase in the solidification strength [15]. Katsioti et al. demonstrated that a solidified agent composed of expansive soil and cement could be used to solidify the heavy-metal-contaminated soil, and the formation of hydrated calcium silicate, calcium hydroxide and ettringite directly led to hardening, thus effectively improving the solidification strength and sealing the heavy metal ions [16]. For the strength and microstructure of solidified soft soil, a lot of research work has been carried out [17]. Li et al. used cement and bentonite to treat sewage sludge and found that cement incorporation should reach a certain level to improve the dry and wet durability of the solidified body [18]. Cha et al. found that the strength of heavy-metal-contaminated soil solidified by cement first increased and then decreased with the increase in the number of dry and wet cycles [19]. Tan et al. used cement and lime to improve silty soil and found that the unconfined compressive strength of the improved silty soil decreased with an increase in the number of freeze–thaw cycles [20]. Fang et al. found that the main factor affecting the compressive performance of soil weakened by the freeze–thaw cycle is water content. Using active MgO as a curing agent [21], Zheng et al. systematically conducted three durability tests on carbonized solidified soil, namely, the dry–wet cycle, freeze–thaw cycle and sulfate attack, and found that it had a good resistance to all three. In addition, to ensure long-term stability, the strength and deformation of solidified soft soil in a complex service environment should not be greatly affected [22,23]. Dong et al. found that the type, size and replacement ratio of waste glass significantly affects concrete durability. Compared to glass cullet, fine glass powder can usually improve long-term durability, because the enhanced pozzolanic reactivity can reduce the ASR expansion due to the densified microstructure and reduced porosity. On the other hand, other factors such as mineral additives and mixing and curing methods also potentially affect durability [24]. Long et al. carried out extensive experiments to analyze the effects of coal gangue on the compressive strength, elastic modulus, stress–strain curve and anti-corrosion of a cement–soil mixture. The results showed that this new clean production of a high-performance cement–soil mixture through waste coal gangue reinforcement has great potential for railway foundation treatments. Therefore, it was very important to systematically study the long-term stability of solidified soft soil.

In previous research, a new solidified agent based on industrial waste residue and recycled fine aggregate was developed, and the formula was obtained, which had a high solidified strength. However, the long-term stability of solidified soft soil was not involved. Therefore, the long-term properties of solidified soft soil, including the immersion test, dry–wet cycle and freeze–thaw cycle, were systematically studied in this paper. Firstly, the immersion stability of solidified soft soil was confirmed, and its difference with that of soft soil without the solidified agent was also clarified. Secondly, the mass and compressive strength loss of solidified soft soil were determined under different numbers of dry–wet cycles, and the microstructure of solidified soft soil was analyzed by scanning electron microscopy (SEM). Finally, the influence of the freeze–thaw cycle on the mass, compressive strength and microstructure of solidified soft soil was confirmed.

## 2. Materials and Methods

### 2.1. Materials

The soft soil was taken from Dongliu road reconstruction and expansion project in Qixia District, Nanjing. According to the relevant requirements in the code for highway geotechnical test (JTG 3430-2020), the particle size, water content, liquid plastic limit and other relevant physical indexes of soft soil were measured. The results are shown in Table 1.

According to the results of previous research, the solidified agent consisted of cement, blast furnace slag and phosphogypsum, with the mass ratio of 1:2.3:0.8.

As a filling material, recycled fine aggregate was directly purchased from the market. Particle size distribution of recycled fine aggregate are shown in Table 2. In addition, the ash content of recycled fine aggregate was 94.4%, and the purity was about 92.9~97.3%.

### 2.2. Experimental Method

The sample was submerged in deionized water. Water molecules immediately infiltrated into the surface pores and surface microcracks of the sample, turning it to a saturated state and weakening the physical and chemical forces between soil particles. Therefore, the curing effect of curing agent can be qualitatively evaluated by comparing the water stability of different solidified silts through immersion test. Soft soil, solidified agent and raw fine aggregate were fully mixed and uniform according to the mass ratio of 1:0.1:0.3, and the mixture was put into a cake-shaped test piece with a diameter of 61.8mm and a height of 20mm. After demolding, the specimen was placed in a curing box with a temperature of 20 ± 2 °C and a humidity greater than 95% for 7 days of curing. Then, a water immersion test was carried out, and the deionized water exceeded the upper surface of the specimen by 20 mm. A professional camera was used to record the changes in the specimen, which lasted for 28 days.

In order to demonstrate the property of solidified soft soil under extreme conditions of drastic changes in temperature and moisture, dry–wet cycle tests were conducted on the soft soil solidified with recycled aggregate and solidified agent, and the mass loss, compressive strength and microstructure change in solidified soft soil were investigated to evaluate the durability. With reference to ASTM D4843-1988, a dry–wet cycle test was carried out. After mixing the soft soil, solidified agent and raw fine aggregate, a cylindrical test piece, with 39.1 mm diameter and 80 mm height, was prepared according to the specification requirement. After demolding, the specimen was placed in a curing box with a temperature of 20 ± 2 °C and a humidity greater than 95% for 28 days of curing. Before the dry–wet cycle test, the initial mass and compressive strength of the specimens were measured. The specimen in the beaker was placed in an oven with a temperature of 60 ± 3 °C for 24 h; then, it was kept in the curing box with a temperature of 20 ± 2 °C and a humidity greater than 95% for 24 h; finally, deionized water was added to the beaker until the specimen was submerged, and the specimen was placed in an environment with a temperature of 20 ± 2 °C for 24 h. The above complete process was recorded as a dry–wet cycle, and different numbers of dry–wet cycles could be realized by repeating the above process.

Due to the seasonal alternation of cold and hot in the northern region, most materials in the project would change to some extent, and solidified soft soil as a building material was no exception. Therefore, it was necessary to study the ability of solidified soft soil to resist freezing and thawing, which was of great significance to explore the service performance under complex climatic conditions. At present, there is no freeze–thaw cycle test specification for solidified soft soil in China. In this paper, freeze–thaw cycle test for solidified soft soil was carried out according to Wang et al.’s study of solidified silt [25]. The specific steps of freeze–thaw cycle test for solidified soft soil were as follows: after soaking the sample in water for 24 h, the initial mass and compressive strength were measured; then, the specimen was put into the freeze–thaw cycle testing machine, the low temperature and high temperature were respectively adjusted to −20 °C and 20 °C, and the automatic temperature inversion time was set to 12 h; after the above freezing and thawing, it was recorded as a freeze–thaw cycle.

### 2.3. Analytical Method

Uniaxial unconfined compressive strength meter was used to measure the strength of solidified soft soil. Three samples in each group were tested in parallel, and the average value was recorded as the unconfined compressive strength. The specific measurement steps were as follows: place the sample on the loading platform; check the sensor to make sure that it is in good contact; press the button to rise to the bottom of the loading platform, so that the top and bottom pressure surfaces are just in contact with the sample; open the program operation interface; set the initial parameters; control the axial strain rate as 1 mm/min. Click the “Start” button for strain rate and then click the “Start Test” button to test. After the test, the sample was taken out, pictures were taken, and the shape of the sample after failure was described.

SEM (FEI Company, Eindhoven, The Netherlands) with a GENESIS 60S energy dispersive X-ray spectroscope (EDS) spectroscopy system with magnification from 500 to 10,000 was used to observe the morphology and to measure the elemental compositions. The accelerating voltage and spot size of the secondary electron detector were 20 kV and 4.0, respectively. The pore structure and the morphology of the products were observed and verified by scanning electron microscopy, and the differences in the morphology of the products were analyzed under different conditions, which provided the basis for revealing the action mechanism of the solidified soft soil.

## 3. Results and Discussion

### 3.1. Influence of Soaking Time on Properties of Solidified Soft Soil

The influence of soaking time on the properties of solidified soft soil is shown Figure 1. It can be seen from Figure 1 that the influence of soaking time on solidified soft soil was obviously different. Figure 1a shows that untreated soft soil showed obvious cracking and destruction for each soaking time. Even if the soaking time was 1 d, cracks occurred in the soft soil. When the soaking time was 1, 3 and 7 d, the cracks showed a gradually increasing trend. However, the crack width seemed to be decreasing when the soaking time was 14 and 28 d. In fact, this did not mean that the amount of damage had decreased. On the contrary, the strengthening of failure led to the disintegration of the solidified soft soil. When the soaking time was 28 d, a large number of soft soil particles were dissolved, which proved that the damage of the solidified soft soil was enhanced. As shown in Figure 1b, the appearance of soft soil solidified with a solidified agent and raw fine aggregate did not change significantly, and it was still intact without damage when the soaking time was increased up to 28 d. Only a few spallings of soil particles were found on the surface of the sample, and no cracks were found. This is because under the comprehensive action of various solidified materials, the overall structure and stability of the silt are enhanced, while the spalling of surface soil particles may be related to the incomplete overall adhesion of the sample during sample preparation. It can be seen that the soft soil solidified with solidified agent and raw fine aggregate had a good water stability, which is a prerequisite for engineering application.

### 3.2. Influence of Number of Dry–Wet Cycles on Properties of Solidified Soft Soil

The influence of the number of dry–wet cycles on the properties of solidified soft soil are shown in Figure 2, Figure 3 and Figure 4. Figure 2 shows the influence of the dry–wet cycle on the mass loss of solidified soft soil. With the increase in the number of dry–wet cycles, the accumulated-mass loss rate of the solidified soft soil gradually increased. When the number of dry–wet cycles was one, three, five and seven, the accumulated-mass loss rate was 1.4%, 3.0%, 4.5% and 6.0%, respectively. The influence of the dry–wet cycle on the compressive-strength loss rate of solidified soft soil is shown in Figure 3. It can be seen from Figure 3 that when the number of dry–wet cycles was one, three, five and seven, the compressive-strength loss rate of solidified soft soil was −10.3%, 13.9%, 41.2% and 53.6%, respectively. When the number of dry–wet cycles was one, the compressive strength of solidified soft soil did not decrease but instead positively increased. The reason was that the cement and blast furnace slag without hydration reacted under immersion and a high–temperature environment, and hydration products with cementitious properties were produced, which improved the compressive strength of solidified soft soil. In addition, the untreated soft soils were destroyed after only one dry–wet cycle, so the untreated soft soils were not compared. When the number of dry–wet cycles was three, the compressive strength of solidified soft soil decreased by 13.9%. With a further increase in the number of dry–wet cycles, the compressive-strength loss rate rose significantly. The main reason for the strength deterioration of solidified soft soil under the condition of dry and wet cycles is that the volume of the sample changes due to the deformation of dry shrinkage and wet expansion, which results in shrinkage and expansion stress and produces fine cracks in the sample. With an increase in the number of dry–wet cycles, cracks continue to develop and connect, resulting in increased water loss and water-absorption degree, which further intensifies the structural damage of the sample caused by the deformation of dry shrinkage and wet expansion, which is manifested in the reduction in sample strength. Thus, a lower number of dry–wet cycles had less impact on solidified soft soil, while a higher number of dry–wet cycles caused serious damage.

The microstructures of solidified soft soil under different curing environments are shown in Figure 4. Figure 4a shows that under the standard curing environment, solidified soft soil was intact as a whole, and its structure was dense, without obvious defects such as cracks and holes. Compared with solidified soft soil under the standard curing environment, solidified soft soil after seven dry–wet cycles showed small cracks, and the structural compactness began to decline (Figure 4b). The synergy of the two aspects, the cementitious minerals produced by the hydration of the solidified agent and the skeletonization from the recycled fine aggregate, improved the mechanical properties and microstructure of the solidified soft soil, which were confirmed in the preliminary work. However, the properties of solidified soft soil declined with the continuous destruction of dry–wet cycles.

### 3.3. Influence of Number of Freeze–Thaw Cycles on Properties of Solidified Soft Soil

The influence of the number of freeze–thaw cycles on the properties of solidified soft soil is shown in Figure 5, Figure 6 and Figure 7. It can be seen from Figure 5 that the accumulated-mass loss rate of solidified soft soil increased gradually with the increase in the number of freeze–thaw cycles. When the number of freeze–thaw cycles was 5, 10, 15 and 20, the accumulated-mass loss rate of solidified soft soil was 12.6%, 16.7%, 17.9% and 18.8%, respectively. In addition, the untreated soft soils were destroyed from only one freeze–thaw cycle, so the untreated soft soils were not compared. During the process of a freeze–thaw cycle, the water on the surface and inside of the solidified soft soil become ice, and the volume expansion compresses the surrounding soil particles. The large amount of expansion stress destroys the cementation between soil particles, resulting in the displacement and even crushing deformation of soil particles. When the frost on the surface and inside of the solidified soft soil melts into water droplets, the water permeates into the structure along the pore or capillary path on the surface of the structure. The migration of water makes the structural elements of the solidified silt soil, such as pore shape and particle arrangement, change significantly. The alternations caused by the freezing and melting of the water on the surface and inside of the solidified soft soil causes the weakening of the structure of the solidified soft soil. In addition, with the increase in the number of cycles, the growth rate of the strength loss rate also increases, which indicates that the strength-weakening rate of the solidified soft soil test samples after a freeze–thaw cycle is also gradually increasing. The above results can also be verified by the compressive-strength loss rate of solidified soft soil. At the beginning of a freeze–thaw cycle, the compressive strength of the solidified soft soil increased slightly, because the cement and blast furnace slag without hydration produced cementitious products. The structure of the material had a decisive influence on its mechanical properties. With the extension of the freeze–thaw cycle, the destruction of the internal structure of the solidified soft soil directly led to the decrease in the compressive strength (Figure 6). Figure 7 shows the microstructure of solidified soft soil after 20 freeze–thaw cycles. There was a crack in the solidified soft soil, and the freeze–thaw resistance was not good. Nevertheless, soft soil with a solidified agent and recycled fine aggregate showed no obvious damage, and the freeze–thaw resistance was relatively superior. After 20 freeze–thaw cycles, the whole soft soil was relatively complete, and the surface and corners fell off less often. Therefore, the solidified soft soil containing a curing agent and recycled fine aggregate has excellent frost resistance and good durability.

## 4. Conclusions

In this paper, the long-term properties of solidified soft soil, including an immersion test, the dry–wet cycle and the freeze–thaw cycle, were systematically studied. Firstly, the immersion stability of solidified soft soil was confirmed, and its difference with that of the soft soil without the solidified agent was also clarified. At any soaking time, cracks in the soft soil without the solidified agent were always present. The appearance of soft soil solidified with a solidified agent and raw fine aggregate did not change significantly, and it was still intact without any damage when the soaking time was increased up to 28 d. Secondly, the mass and compressive strength loss of solidified soft soil were determined under different numbers of dry–wet cycles, and the microstructure of solidified soft soil was analyzed by scanning electron microscopy. When the number of dry–wet cycles was one, three, five and seven, the accumulated mass loss rate was 1.4%, 3.0%, 4.5% and 6.0%, respectively, and the compressive strength loss rate was −10.3%, 13.9%, 41.2% and 53.6%, respectively. Compared with solidified soft soil under the standard curing environment, solidified soft soil after seven dry–wet cycles showed small cracks, and the structural compactness began to decline. The synergy of the two aspects, the cementitious minerals produced by the hydration of the solidified agent and the skeletonization from the recycled fine aggregate, improved the mechanical properties and microstructure of the solidified soft soil. However, the properties of the solidified soft soil declined with the continuous destruction of the dry–wet cycles. Finally, the influence of the number of freeze–thaw cycles on the mass, compressive strength and microstructure of solidified soft soil was confirmed. When the number of freeze–thaw cycles was 5, 10, 15 and 20, the accumulate mass loss rate was 12.6%, 16.7%, 17.9% and 18.8%, respectively. The microstructure of the solidified soft soil was damaged, and the increase in porosity was the main reason for its strength reduction or even failure. Nevertheless, soft soil with a solidified agent and recycled fine aggregate showed no obvious damage to the microstructure, and the freeze–thaw resistance was relatively superior. The long-term properties of the soft soil solidified with industrial waste residue and regenerated fine aggregate was verified, which provides a foundation for engineering application.

## Figures and Tables

**Figure 1 materials-16-02447-f001:**
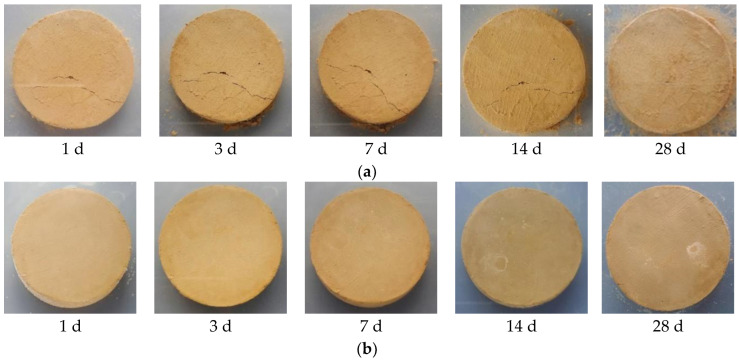
Influence of soaking time on properties of solidified soft soil: (**a**) untreated soft soil and (**b**) soft soil solidified with solidified agent and raw fine aggregate.

**Figure 2 materials-16-02447-f002:**
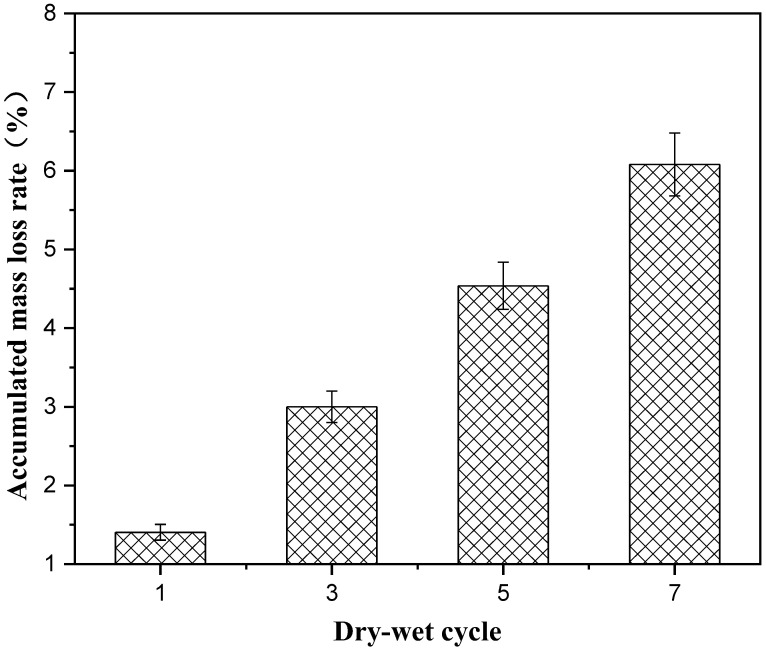
Influence of the number of dry–wet cycles on accumulated-mass loss rate of solidified soft soil.

**Figure 3 materials-16-02447-f003:**
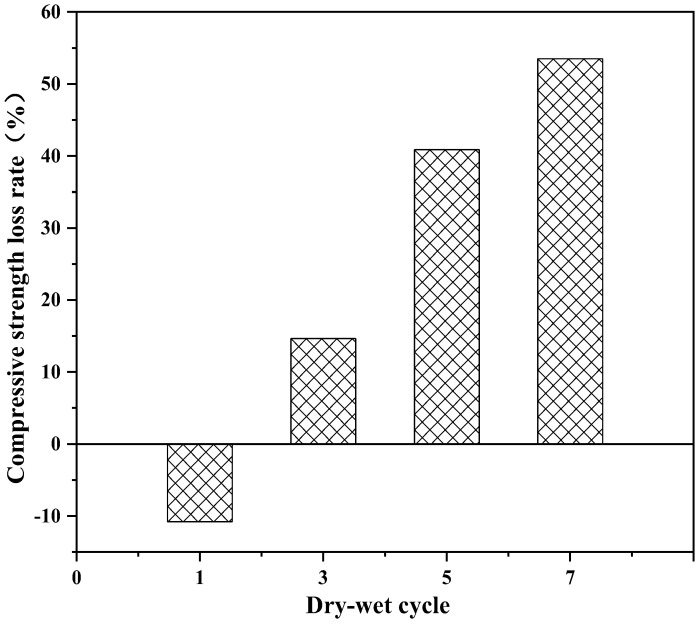
Influence of the number of dry–wet cycles on compressive-strength loss rate of solidified soft soil.

**Figure 4 materials-16-02447-f004:**
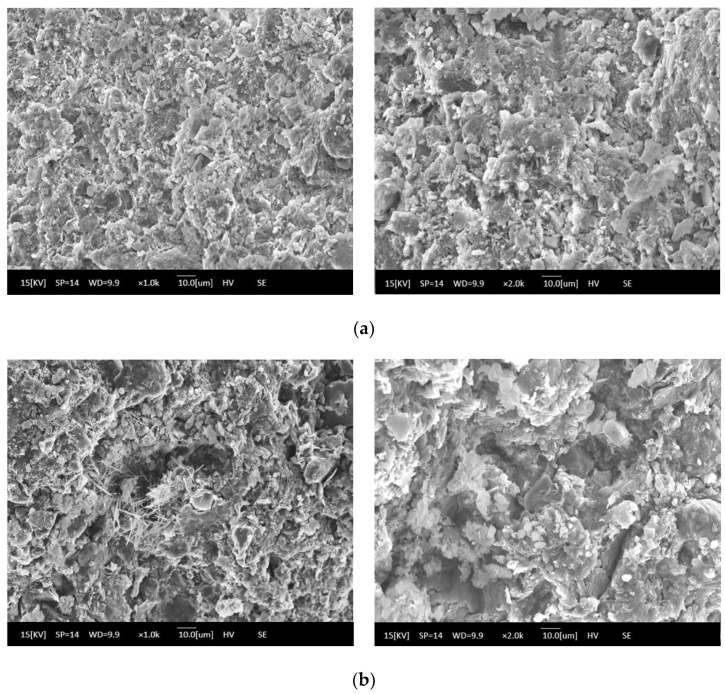
Microstructure of solidified soft soil under different curing environments: (**a**) standard curing and (**b**) 7 dry–wet cycles.

**Figure 5 materials-16-02447-f005:**
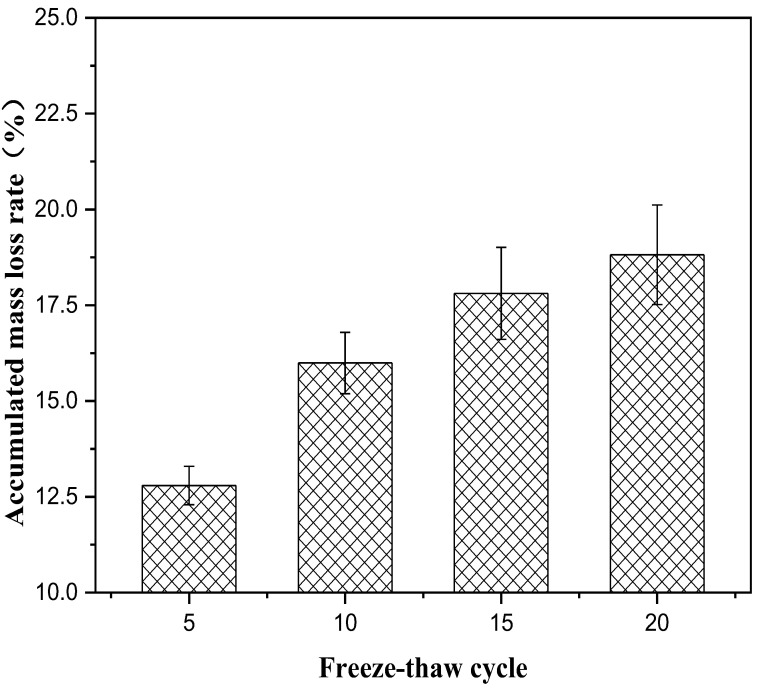
Influence of the number of freeze–thaw cycles on accumulated-mass loss rate of solidified soft soil.

**Figure 6 materials-16-02447-f006:**
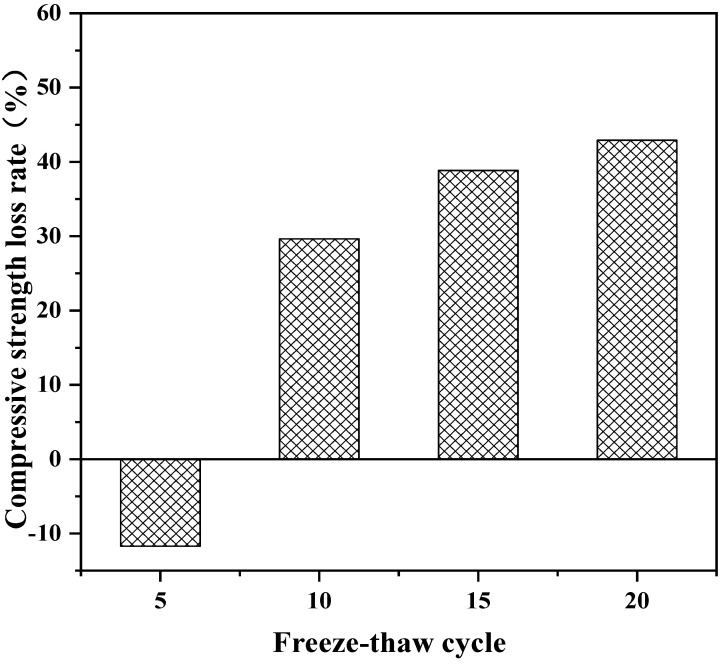
Influence of the number of freeze–thaw cycles on compressive-strength loss rate of solidified soft soil.

**Figure 7 materials-16-02447-f007:**
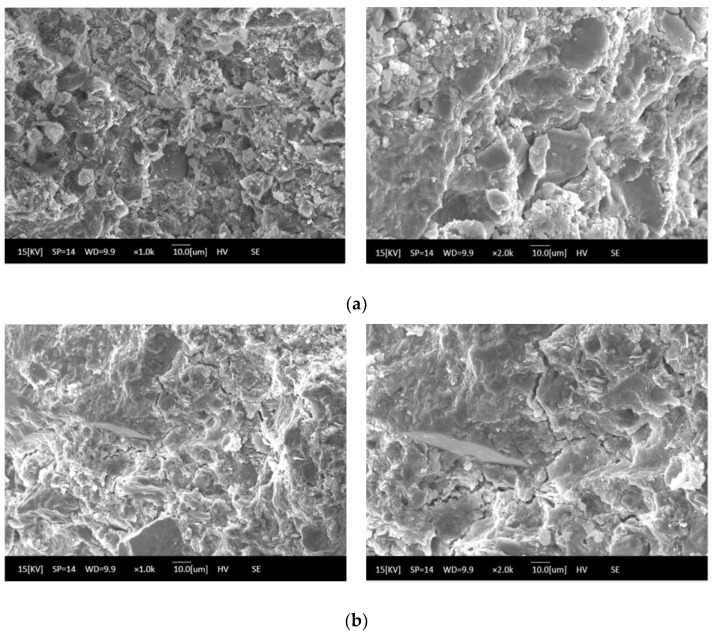
Microstructure of solidified soft soil under different curing environments: (**a**) standard curing and (**b**) 20 freeze–thaw cycles.

**Table 1 materials-16-02447-t001:** Main physical indexes.

Water Content (%)	Liquid Limit (%)	Plastic Limit (%)	Plasticity Index	Maximum Dry Density (g/cm^3^)	Clay Content (%)	Organic Matter Content (%)	pH
45.6	37.2	22.9	14.3	1.82	46	1.72	6.5

**Table 2 materials-16-02447-t002:** Particle size distribution of recycled fine aggregate.

Aperture (mm)	Residue (g)	Proportion (%)	Proportion of Accumulated Residue (%)
10.00	0	0	0
5.00	130.77	6.54	6.54
2.50	527.00	26.35	32.89
1.25	313.45	15.67	48.56
0.630	315.37	15.77	64.33
0.315	245.47	12.27	76.60
0.160	213.64	10.68	87.28
0	254.30	12.72	100

## Data Availability

Not applicable.

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
