# Peer review of "Study on Long Term Property of Soft Soil Solidified with Industrial Waste Residue and Regenerated Fine Aggregate"

_materials, 2023, doi:10.3390/ma16062447_

Round 1
Reviewer 1 Report
Background:
The long-term property of solidified soft soil, including immersion test, dry-wet cycle and freeze-thaw cycle, were systematically studied. Firstly, the immersion stability of solidified soft soil was confirmed. The appearance of soft soil solidified with solidified agent and raw fine aggregate had not changed significantly, and it was still intact without damage when the soaking time was up to 28 d. Secondly, the mass and compressive strength loss of solidified soft soil were determined.
General comment: This is an important topic relevant to the journal and to the readership. The methodology is sound, the results are generalizable, and add significantly to what is already known about this topic. There are no problems with ethics or conflict of interest." and should be accepted in the present form.
Abstract:
In the abstract, the study conclusion is not mentioned.
Introduction:
I have no comments; very well written, and in the introduction needed to add the recent references, there is plenty of similar studies related to organic wastes amendments, etc. (please see below that you can consult with their reference lists), that need to be more connected with the specific problem of this study.
https://doi.org/10.1016/j.jksus.2022.102212
https://doi.org/10.1080/00103624.2022.2071930
- https://doi.org/10.1080/01904167.2020.1849293
General questions:
1- Do you measure the effect of organic wastes soil holding capacity? (It is a very important soil parameter)
2- You measured the physical properties so soil, what are the main chemical characteristics of the soil?
3-Do the authors need to explain why they choose the selected soil type?
Results & Discussion:
very well written.
Conclusion
In your conclusion, please discuss limitations of the present study and future research to address them.
References
The references very little only 16 references, please add some references.
Thank you
Author Response
1、In the abstract, the study conclusion is not mentioned.
Response:
According to the reviewer's comments, the abstract has been carefully processed, and the results are analyzed in depth.
2、Do you measure the effect of organic wastes soil holding capacity? (It is a very important soil parameter)
Response:
The reviewer's comments are highly appreciated, the effect of organic wastes soil holding capacity is indeed a very important parameter for soil, and we have also done relevant tests. but this paper is mainly aimed at the long-term property of soil, so the relevant results did not added.
3、You measured the physical properties so soil, what are the main chemical characteristics of the soil?
Response:
In addition to the physical properties of the soil, the content of organic matter and pH of the soil are also tested. In addition, the main chemical composition of the soil is silica, so there is no further discussion.
4、Do the authors need to explain why they choose the selected soil type?
Response:
The goal of the test is that the cured soil can be used as a roadbed, so soft soil is a good choice. The regional difference of soil properties is too large, so it is necessary to carry out extensive research on the technology of curing soft soil.
5、The references very little only 16 references, please add some references
Response:
According to the reviewer's comments, research status has been supplemented. The references have been greatly increased to 21.

Reviewer 2 Report
I read manuscript and found followings deficiencies, need to be addressed in final draft:
· The rational statement, describe the need/importance of research is missing in abstract.
· An abbreviation could be better representative for “solidified soft soil” i.e. SS soil.
· The abstract should be self-explanatory. The methodologies need to be given in abstract.
· There is transitional gap between paragraphs especially in Introduction section. More reviews need tom be incorporated in this section to describe the hypothesis, objectives and need of the study.
· It is better to show the results (Table 1) in Result section.
· The methodology section is not written well. The description of “water immersion test” and “dry-wet cycle test” should be incorporated.
· A comprehensive flow chard of processes and schematic diagrams could be better representative for readers.
· There is no discussion in results section. A separate section of Discussion needs to be added and support your results with references.
· The conclusions are too lengthy, please summarize conclusions as objectives of the study.
· The manuscript is simple and need to be improved as per standards of highly impacted journal “Materials”.
· Good luck
Author Response
1、 An abbreviation could be better representative for “solidified soft soil” i.e. SS soil.
Response:
According to the reviewer's comments, it has been revised.
2、The abstract should be self-explanatory. The methodologies need to be given in abstract.
Response:
According to the reviewer's comments, it has been revised.
3、There is transitional gap between paragraphs especially in Introduction section. More reviews need tom be incorporated in this section to describe the hypothesis, objectives and need of the study.
Response:
According to the reviewer's comments, the introduction has been improved. The hypothesis, objectives and need of the study have also been added.
4、 The methodology section is not written well. The description of “water immersion test” and “dry-wet cycle test” should be incorporated.
Response:
According to the reviewer's comments, the description of “water immersion test” and “dry-wet cycle test” have been supplemented in methodology section.

Reviewer 3 Report
Valuable research has been done This research is about Study on long term property of soft soil solidified with indus- 2trial waste residue and regenerated fine aggregate. The long-term property of solidified soft soil, including immersion test, dry-wet cycle and freeze-thaw cycle, were systematically studied. Firstly, the immersion stability of solidified soft soil was confirmed. The appearance of soft soil solidified with solidified agent and raw fine aggregate had not changed significantly, and it was still intact without damage when the soaking time was up to 28 d. Secondly, the mass and compressive strength loss of solidified soft soil were determined. When the dry-wet cycle was 1, 3, 5 and 7, the accumulated mass loss rate was 1.4%, 3.0%, 4.5% and 6.0% respectively, and the compressive strength loss rate was -10.3%, 13.9%, 41.2% and 53.6% respectively. Compared with soft soil solidified without any material, soft soil solidified with industrial waste residue and raw fine aggregate was dense on microstructure without obvious damage. Finally, the influence of freeze-thaw cycle on the mass, compressive strength and microstructure of solidified soft soil was confirmed. When the freeze-thaw cycle was 5,10,15 and 20, the accumulate mass loss rate was 12.6%, 16.7%, 17.9% and 18.8% respectively. Nevertheless, soft soil with solidified agent and recycled fine aggregate had no obvious damage on microstructure, and the freeze-thaw resistance was relatively superior.
In my opinion this paper can be accepted after major revision and re-evaluation.
The following corrections are recommended:
It is recommended to provide more complete explanations regarding research innovation and the need to conduct research.In the microscopic photos, the scale is not clear. It is suggested to be corrected.
Figure 1 is a good figure. Provide more explanations about this form.It is suggested to provide more complete explanations in the Analytical method section
It is recommended to use newer references and delete old references.
Author Response
1、 It is recommended to provide more complete explanations regarding research innovation and the need to conduct research.
Response:
According to the reviewer's comments, the research ideas and innovations of this paper have been supplemented in research status.
2、In the microscopic photos, the scale is not clear. It is suggested to be corrected.
Response:
According to the reviewer's comments, SEM images has been carefully processed, and the scale is clearly visible.
3、Figure 1 is a good figure. Provide more explanations about this form.
Response:
According to the reviewer's comments, it has been revised.
4、It is suggested to provide more complete explanations in the Analytical method section
Response:
According to the reviewer's comments, it has been revised.
5、It is recommended to use newer references and delete old references.
Response:
According to the reviewer's comments, it has been revised.
Reviewer 4 Report
The manuscript must be modified before publication, some comments are listed below:
1. 'soft soil solidified without any material' should be corrected to untreated soft soil or pristine soft soil. Please double-check the whole manuscript.
2. Another paragraph in Introduction is recommended to add. Please consider simply introducing the industrial wastes and recycled aggregate for construction: Resources, Conservation and Recycling, 2021, 172: 105664. Journal of cleaner production, 2019, 231: 468-482. Journal of Cleaner Production, 2021, 314: 127968.
3. Other properties of recycled fine aggregate are recommended to present, including ash content and purity.
4. When you mention no standard for the freeze-thaw cycles, do any other references mention the freeze-thaw cycle test on the soft soil? I would suggest considering the previously used method, and then you are able to compare the results.
5. Section 3.1, the first paragraph can be deleted or moved to the section of experimental preparation. 'Influence of soaking time on property of solidified soft soil was shown Fig.1. ' Move the sentence to the second paragraph.
6. Section 3.2, same comment to the previous one, and remove or move the sentences. It is not related to your results and discussion.
7. For better comparison, the Fig. 2, 3, 4, 5, 6 should include the properties of untreateed soft soil as well. I would suggest author consider adding this part of the results to improve the article's quality.
Author Response
1、'soft soil solidified without any material' should be corrected to untreated soft soil or pristine soft soil. Please double-check the whole manuscript.
Response:
According to the reviewer's comments, the reason why untreated soft soil cannot undergo wet-dry or freeze-thaw cycles is that it will be destroyed after one cycle. Therefore, no comparison has been made on untreated soft soil.
2、Another paragraph in Introduction is recommended to add. Please consider simply introducing the industrial wastes and recycled aggregate for construction: Resources, Conservation and Recycling, 2021, 172: 105664. Journal of cleaner production, 2019, 231: 468-482. Journal of Cleaner Production, 2021, 314: 127968.
Response:
According to the reviewer's comments, the introduction has been greatly improved by reading the literatures recommended by reviewers.
3、Other properties of recycled fine aggregate are recommended to present, including ash content and purity.
Response:
The particle size distribution of recycled fine aggregate is very important, but its composition and product quality are difficult to be measured by some indicators. Subsequent tests will carefully consider the suggestions of reviewers.
4、When you mention no standard for the freeze-thaw cycles, do any other references mention the freeze-thaw cycle test on the soft soil? I would suggest considering the previously used method, and then you are able to compare the results.
Response:
According to the reviewer's comments, it has been revised.
5、Section 3.1, the first paragraph can be deleted or moved to the section of experimental preparation. 'Influence of soaking time on property of solidified soft soil was shown Fig.1. ' Move the sentence to the second paragraph.
Response:
According to the reviewer's comments, it has been revised.
6、Section 3.2, same comment to the previous one, and remove or move the sentences. It is not related to your results and discussion.
Response:
According to the reviewer's comments, it has been revised.
7、For better comparison, the Fig. 2, 3, 4, 5, 6 should include the properties of untreated soft soil as well. I would suggest author consider adding this part of the results to improve the article's quality.
Response:
According to the reviewer's comments, the reason why untreated soft soil cannot undergo wet-dry or freeze-thaw cycles is that it will be destroyed after one cycle. Therefore, no comparison has been made on untreated soft soil.

Round 2
Reviewer 2 Report
Good Luck!
Author Response
accepted
Reviewer 3 Report
The desired corrections have been made. In my opinion, the article can be accepted.
Author Response
accepted
Reviewer 4 Report
The comments from reviewers are not entirely solved. For comment 1, I mean the soil solidified without any materials can be written as untreated or pristine soft soil, authors did not get my point. Comments 3 and 4 are also not well explained. For comment 7, if the untreated soil was really bad, the authors need to mention this situation both in the content and figures, but I did not see these.
Author Response
- 'soft soil solidified without any material' should be corrected to untreated soft soil or pristine soft soil. Please double-check the whole manuscript.
According to the reviewer's comments, it has been revised.
- Another paragraph in Introduction is recommended to add. Please consider simply introducing the industrial wastes and recycled aggregate for construction: Resources, Conservation and Recycling, 2021, 172: 105664. Journal of cleaner production, 2019, 231: 468-482. Journal of Cleaner Production, 2021, 314: 127968.
According to the reviewer's comments, the literature recommended by the reviewer was reasonably quoted and modified in the Introduction.
- Other properties of recycled fine aggregate are recommended to present, including ash content and purity.
According to the reviewer's comments, it has been revised.
- When you mention no standard for the freeze-thaw cycles, do any other references mention the freeze-thaw cycle test on the soft soil? I would suggest considering the previously used method, and then you are able to compare the results.
According to the reviewer's comments, this paper has referred to the freeze-thaw cycle in other literatures and modified this part. This author is an authority in geotechnical engineering for reference.
- Section 3.1, the first paragraph can be deleted or moved to the section of experimental preparation. 'Influence of soaking time on property of solidified soft soil was shown Fig.1. ' Move the sentence to the second paragraph.
According to the reviewer's comments, it has been revised.
- Section 3.2, same comment to the previous one, and remove or move the sentences. It is not related to your results and discussion.
According to the reviewer's comments, it has been revised.
- For better comparison, the Fig. 2, 3, 4, 5, 6 should include the properties of untreated soft soil as well. I would suggest author consider adding this part of the results to improve the article's quality.
According to the reviewer's comments, it has been revised and mentioned in the content.
